# Promoting Physical Activity and Health in the Workplace: A Qualitative Study among University Workers, Spain

**DOI:** 10.3390/ijerph20032350

**Published:** 2023-01-28

**Authors:** Antonio Jesús Casimiro-Andújar, Juan Carlos Checa, María-Jesús Lirola, Eva Artés-Rodríguez

**Affiliations:** 1Department of Education, Faculty of Education Sciences, University of Almería, 04120 Almería, Spain; 2SPORT Research Group (CTS-1024), CERNEP Research Center, University of Almería, 04120 Almería, Spain; 3Department of Geography, History and Humanities, Faculty of Humanities, University of Almería, 04120 Almería, Spain; 4Department of Psychology, Faculty of Psychology, University of Almería, 04120 Almería, Spain; 5Area of Statistics and Operative Research, Department of Mathematics, Faculty of Sciences, University of Almería, 04120 Almería, Spain

**Keywords:** health promotion, UAL Activa, sedentary lifestyle, healthy habits

## Abstract

The social changes we have been experiencing in recent years are generating anxiety that, together with sedentary lifestyles and poor health habits, are leading to premature ageing of society, in addition to the high rates of obesity and associated morbidity. In order to improve the lifestyles of the university community, the objective of this research project was the implementation of the UAL Activa programme based on physical activity. The beneficiaries were UAL staff from different sections, a total of 68 participants aged between 28 and 61 years (*M_age_* = 49.36). Additionally, a total of 12 final year students of the degree in physical activity and sport sciences were responsible for designing and developing the exercise tasks. In-depth interviews were held with the participants of this project about the different benefits that their participation had brought them. The results obtained from the analysis with Nvivo v.10. showed five thematic blocks on the perceived improvements in physical fitness, physical condition, mood and emotional state, assessment of the new lifestyle, and social relations. In conclusion, the benefits of physical activity and the need to continue implementing action and intervention plans to encourage and promote its practice in all sectors of the population were highlighted.

## 1. Introduction

The world is currently experiencing an epidemic of sedentary lifestyles caused by a “robotisation” of play, mechanisation of work, automation of tasks, “concretisation” of the environment, as well as an abuse of passive technological leisure [1,2,3]. All this leads to a decrease in the level of physical activity and a decrease in physical capacity, which results in an increase in early morbidity and mortality. Hence, one of the pioneers of sports medicine in Spain [4] indicates that more than half of the population does not die, but rather elegantly “commits suicide”, basing their lives, on the one hand, on five “too many” (alcohol, tobacco, food, stress, and drugs) and, on the other hand, on “too little” exercise. In other words, the lifestyle in postmodern societies is characterised, among other things, by a sedentary lifestyle, inactivity, and a lot of rushing around [5,6]. To the extent that the most sedentary people tend to suffer from low-grade inflammatory processes and have greater difficulty in proper blood and lymphatic circulation, which oxygenates all tissues and purifies toxins, so they are more likely to suffer a cardiovascular accident and other diseases typical of a hypokinetic society [7].

Indeed, most of the adult population in Western countries suffers from a high degree of stress. The social changes we have experienced in recent years are generating a lot of anxiety, suffering in the present for something that we do not know for sure will manifest itself in the future. My life was full of misfortunes, many of which never happened, stated Descartes.

Moreover, the prevalence of chronic anxiety is increasing due to fast-paced lifestyles, toxic competition, job and economic uncertainty, the recent global coronavirus pandemic, etc., leading to insomnia, lack of concentration, and nervousness, making it imperative to face fear and to move forward with acceptance, determination, and personal and social responsibility.

The aforementioned anxiety usually goes hand-in-hand with excessive stress, which is produced by a sustained situation in which the demands exceed the natural capacity for self-regulation, so that the stimulus disturbs the balance or homeostasis, causing reactivity and nervousness (muscle tension, agitated breathing, etc.) [8]. Initially, any stressful situation helps us to get out of vital emergency situations and improve performance, but when stress is intense and chronic, the process is reversed and produces harmful effects on health: insomnia, back pain, digestive and cardiovascular problems, infections, sterility, sexual impotence, irritability, and decreased concentration, while also affecting social relationships [9,10].

Therefore, in order to provide our life with integral well-being and better manage chronic stress, we must, among other things, do pleasant physical exercise, relaxation, meditation, socialisation with tonic—not toxic—people, laugh, and have a good mood and optimistic, positive, and grateful thoughts.

Thus, physical-sports practice is an ideal tool to *soften* and *coexist* with the aforementioned chronic stress [11,12,13]. Anyone who habitually practices pleasurable exercise knows the feeling of well-being that accompanies its completion, improving and exercising mental function and having a euphoriant effect, as a result of the release of endorphins. Furthermore, if this practice is in accordance with the interests and abilities of the practitioner, it produces an increase in self-esteem, creativity, willpower, attention, concentration, perseverance, self-knowledge, commitment, self-control, self-confidence, greater cognitive reserve, etc., helping to prevent neurodegenerative diseases over the years.

Therefore, physical exercise becomes a key factor for psychophysical and emotional balance, energy contribution, and exceptional development of the holistic health of the person. The benefits of physical exercise are many and varied. From the individual point of view, it facilitates adaptation to effort, stimulates the desire to excel, courage, discipline, mental strength, etc., and, of course, promotes integral health [14,15,16]. In its collective form, it induces teamwork, improves interpersonal relationships, the acceptance of rules, etc. All this favours the education of individual and social values so important in our society, such as tolerance, cooperation, respect, solidarity, attention to diversity, co-education, etc. [17,18,19].

This is why, nowadays, there is a boom of fitness practitioners, popular runners, triathletes, marathon runners, cyclists or walkers, exercising in gyms, promenades, parks, and mountains [20,21]. However, the increase in this practice has not been paralleled by an increase in physical culture, which unfortunately leads to some “sporting” exercises of dubious healthiness, with significant health risks. Therefore, it is not a question of moving much more, but of moving better, since when it is not practised in a controlled and progressive manner, it can be very harmful [22,23,24].

Following the above, a moderate physical practice planned by good specialists is possibly the best prevention and an exceptional medicine for most chronic diseases, being the perfect tool for holistic health, understood as a state of homeostasis, integral well-being, vitality, and inner serenity, as it allows the integral development of its four dimensions:Physically: cardiovascular, respiratory, metabolic, muscular, etc., benefits.Mentally: improved neuroplasticity, concentration, attention, performance, cognitive reserve, etc.Emotionally: improves mood, interpersonal relationships, etc.Spiritually: it acts as an inner pilgrimage towards personal development: self-knowledge, serenity, etc.

Basically, there is no other element that can favour the balanced development of the four planes in interaction and simultaneously. Physical exercise should be a natural part of every person’s daily life, as it favours their integral development, both on a biological, psycho-emotional, and spiritual level. There is no excuse, you have to *get active with common sense*.

Nevertheless, and despite all the above, even the Dalai Lama noted that what surprised him most about humanity “is the man (or woman), because he sacrifices his health to earn money and, when he succeeds, he sacrifices his money to regain his health. He is so anxious about the future that he does not enjoy the present; the result is that he lives neither the present nor the future; he lives as if he would never die, and then he dies without ever really having lived”.

For this reason, we are convinced that if the effectiveness of the human factor is the cornerstone for any company or institution, physical activity (PA) is no less so for these workers, since it has a positive influence both on the physical aspect through health, and on the effectiveness of the job, evaluated from the point of view of profitability. Hence, there is a need for health promotion in the workplace, improving the quality of life of employees and managers, fostering *healthy* interpersonal relationships among all colleagues, as well as reducing absenteeism and improving the working environment [25,26,27].

Sedentary lifestyles and physical inactivity increase the frequency and duration of work incapacity (sick leave), due to a structural and functional deterioration of the organism, which results in the appearance of various diseases, mainly in the musculoskeletal area (back pain and psychological) (anxiety and work stress). For this reason, ergonomics, adequate movement, and postural hygiene at work must be encouraged through compensatory physical exercise to reduce intradiscal pressure [28,29].

However, one of the keys to favouring the promotion of physical exercise in the workplace is the possibility of recharging workers’ inner energy. If we do not have energy, it is very difficult for us to give light to our colleagues, clients (or students in the case of university students), and to be able to efficiently face our work obligations. Undoubtedly, the master key is the combination of meditation, healthy nutrition, emotional development, and adequate physical exercise. All of this can be offered and stimulated in the workplace through holistic health programmes, thus allowing us to increase this energy in the different levels of the human being [30,31]. In this way, the person will be more balanced, favouring the sum of active and healthy employees to achieve a truly healthy company.

All this justification leads us to a double objective: on the one hand, the design of a personalised training programme called UAL Activa, offered by the University of Almeria (UAL) with the aim of improving the physical condition and overall health of the university community, in order to achieve a more active and healthy university. Additionally, on the other hand, to know the opinion of the users of the UAL Activa programme, once their annual training plan has been completed.

Thus, the aim of this research project was the implementation of the UAL Activa programme and its evaluation through qualitative research. For this purpose, information was collected from a sample of workers who participated in the UAL Activa programme, regarding their experiences in the programme, by investigating their personal accounts and opinions, as well as written feedback on their experiences.

## 2. Method

### 2.1. Study Design

This study used the qualitative method and the conventional content analysis approach to understand the impact of participation in the UAL Activa project by employees of the University of Almeria. Qualitative content analysis is a robust process that was used to analyse textual data and aimed to initiate the introductory meanings of the concept. It is also a systematic method of categorization and coding. The main objective of this method is to satisfactorily understand the phenomenon under study [32]. As qualitative research, this study conducted in-depth interviews with individuals to better understand and gain richer insights into participants’ experiences and perspectives [33].

### 2.2. Context and Study Settings

At the University of Almeria (UAL), we are developing an ambitious and innovative project offered from the Sports Unit of the UAL and aimed at all interested teaching and research staff and administrative and service staff, in order to improve healthy habits, body composition, physical condition, postural hygiene, quality of life, reduce anxiety and work stress, and increase the level of PA of the same through an initial diagnosis, planning and individualised programming, and a final evaluation. All of this is carried out by virtue of an internal service provision directed by a specialist teacher of PA for health and executed by final year students in physical activity and sport sciences of the University of Almeria.

This is a service of the Sports Unit for UAL professionals, which consists of an individualised physical programme for each participant, with the advice, programming, and monitoring of a final year student of the degree in physical activity and sport sciences of the University of Almeria. This person must assess the level of physical activity, physical condition, body composition, and initial and final quality of life of each assigned participant, ensuring the correct technique and execution of the exercises, prioritising the health and well-being of each person.

In addition, they must act as a coach for the coachee to accompany them in achieving other personal objectives related to quality of life (nutrition, rest, stress management, social relations, etc.), launching weekly challenges to promote a healthy lifestyle (conscious breathing, weight loss, fruit consumption, active transport, etc.).

UAL-Activa is a theoretical-practical learning programme for Sport Science students, while offering a supervised exercise service aimed at improving the health and quality of life of all those enrolled, in an attempt to achieve a more active and healthy university. The main objective is to promote the transfer of knowledge to the university community and develop learning spaces where the participating students take an active part in their learning, as well as receiving a specific training programme and tutored monitoring throughout the process.

This programme, which has been running in the university’s fitness room since January 2018, has been very well received by the university community. The university students involved work on a personalised basis with each enrolled worker, who undergoes an initial assessment in which they are asked about their interests, availability, and objectives for their participation in the programme.

The tasks are developed according to the objectives and level of the user, two 60-min sessions per week for 6 months. The tasks developed in each session were mainly of strength through self-loading, elastic bands, weight machines, TRX, and free weights, through various movement patterns (horizontal and vertical pushes and pulls, hip and knee dominant exercises, as well as development of the body’s stabilising musculature (core)). Tasks are also carried out to improve cardiorespiratory capacity, flexibility, breathing, and relaxation, all with exhaustive control and a personalised progression of the intensity of the load.

### 2.3. Participants

The programme was open to all university employees who had no medical contraindications for exercising. A total of 68 members from the administration-services staff and teaching-research staff sections of the University of Almeria (36 men; 32 women) aged between 28 and 61 years (*M_age_* = 49.36) participated in the programme, which lasted six months. A total of 12 students—volunteers and paid—from the last year of the degree in physical activity and sport sciences were responsible for designing and developing the physical exercise (PE) tasks, always supervised by a PhD professor or a specialist in physical activity and health from that university.

The recruitment strategy was to disseminate the exercise programme UAL Activa through the sports service website, internal email for the entire university community and social networks, and through information leaflets (see Figure 1).

The staff interested in participating had to sign a commitment to participate and adhere to the programme, which started with an initial interview, in which the workers were asked to provide the following information:Name.Daily duties at work.Method of transport.Perceived level of stress.Hours of sleep.Usual type of food.Current pain/injuries.Do you take medication regularly and for what?Current physical exercise/sport.Previous physical exercise/sport.Preferred type of activity you would like to do.Preferred leisure activities in your spare time.Aim of the programme.Degree of involvement.

### 2.4. Data Collection Methods

After obtaining approval from the Ethics Committee of the University of Almeria, data were collected through in-depth interviews with the participants of the UAL Activa project. The interviews were conducted in a welcoming and familiar environment for the participants. The interviewees were informed about the ethics of confidentiality of the information and their right to leave the interviews at any time. In addition, for the interviews, all participants gave their consent and all participants’ consent. None of the participants refused to participate in the research after learning about the research objectives and the need for future aspects.

For the coachees’ evaluation of their experiences in the programme, a semi-structured interview was used in which they were asked about the most important aspects of the programme, their degree of involvement, possible continuity, and their opinions about possible improvements in the event of repeating the experience. Participants’ responses to these questions were requested in handwritten form.

### 2.5. Data Management and Analysis

To assess the quality of the study, Lincoln and Guba’s reliability criteria were followed [34]. Researchers participated in data collection for 6 months and completed data collection in different ways (triangulation). A qualitative research expert and a sociologist supervised the research to check the accuracy of the data and analysis (expert review), and data were given to some of the participants to confirm accuracy and correct if necessary (member checking). Maximum variation in sampling was ensured to obtain more complete information (diversity in sampling by conducting interviews with all participants). Throughout the process of this research and dissemination of the results, the researchers tried to follow the criteria proposed by Tong et al. (COREQ) that are used for the results of qualitative studies [35]. Finally, the responses collected were used to carry out a syntactic analysis of the responses and to extract the five main categories that would appear in their discourses using the Nvivo v.10 programme.

## 3. Results

### User Evaluation of the UAL Activa Programme

In this section, the main results from in-depth interviews with those users who voluntarily wished to participate in the research are shown. It should be noted that in order to preserve the anonymity of the participants, we will only insert the initials of their names. Table 1 summarizes the categories and sub-categories derived from the qualitative analysis.

Consequently, as we pointed out above, the main purpose of the UAL Activa project is to improve the physical condition and health status of the participants in the different spheres or dimensions. From the physical point of view, the programme has been a success as stated by the participants who voluntarily took part in guided interviews and provided feedback on the programme, as in the case of their **physical health**:


*“I feel much better than I did 6 months ago...; with the entry into the UAL Activa programme I started to see substantial improvements, from the physical and mental point of view, etc.”*
(C.C.)


*“So, I insist, better than before. And with a blood test that gave me some great results.”*
(J.L.)


*“Of course, in the last few months, when I was playing sports, I used to feel pain in my knees. Since I started the UAL Activa sessions I have changed some of the habits that aggravated this pain, it doesn’t hurt as much and my legs feel stronger, which means that my physical condition is changing. Also, I sleep better now.”*
(C.C.)

When delving into the **fitness** gains achieved by the UAL Activa step, these differ from participant to participant:


*“If it is flexibility in my case, as I have lost weight I can now tie my shoelaces without having to bend down too much.”*
(J.T.)


*“In my case, endurance. I like to take the bike from time to time and I’ve managed to extend the stages.”*
(J.A.S.)


*“The one I’m better at is strength. I was good before and now I’m... even better.”*
(C.C.)

If the achievements of the programme in health and physical condition have been remarkable, the participants highlight mainly the improvement in the **mood and emotional state**, pointing out aspects such as optimism, vitality, positive outlook on daily challenges, or greater relaxation, as reflected in the following testimonies:


*“In my case better than before. I have felt much more positive in life. Since I’ve been doing sport, I feel much more positive, happy and, above all, practical (...) more optimistic and vital, it’s very clear to me.”*
(J.A.S.)


*“Well, I’ve had a better time this year. The stress at the end of the school year, which is absolutely crazy, well, I can say that I’ve handled it much better this year, with more energy.”*
(T.L.)


*“In my case it has influenced me in such a way that, when I had to do a session, you put aside everything you had to do, you came with the desire and joy of doing the activity. And at the end I leave with the satisfaction of doing something that was positive for me.”*
(C.C.)


*“It has made me burn energy and helps me relax.”*
(T.A.)

They also manifest themselves in other **lifestyle values** such as discipline or daily habit, which go beyond the physical or mental health aspects:


*“For me the positive thing has been, for example, the fact of establishing a habit. Going every week with the coach or trainer... for me it has been fundamental. Physically it has helped me to de-stress, it has taken away a lot of back pain and it has helped me to perform well and therefore to rest.”*
(T.L.)


*“At weekends, I do go for walks, hiking, hikes, uphill, etc. And then, nowadays I try to go for a walk whenever I can and never take the lift and try to walk in the mornings.”*
(J.L.)


*“I have a motorbike and I haven’t ridden it for a month and a half. Now I go everywhere with my bike. Apart from coming here to the university, any other errands I had to do, I used to take the bike and now the truth is that I don’t even think about it.”*
(T.L.)


*“On a day-to-day basis, I do move around quite a lot. If it’s no more than a fourth floor, I take the stairs, not the lift; unless it’s more than 2 kilometres and I’m in a hurry, I walk instead of taking the car; then, in classes I never sit down and, even if I’m working on the computer, it’s rare that I don’t get up. Also, on a normal day, I do cross the campus five or six times perfectly well.”*
(T.A.)

Even the value of the gym as a space, familiarisation with equipment or machines, and, above all, as a place for **social interaction** with co-workers:


*“The same thing happened to me. I had never been to a gym in my life and as a result of these sessions, I can’t say that I love going to the gym, but being advised by someone really professional and even considering him a friend has made me value it in a much more positive way. It has also encouraged me, for example, to take up cycling much more regularly; in fact, I am now cycling to work. So sport is now, for me, a part of my life.”*
(T.A.)


*“It has given me mental strength. I thought I couldn’t be able to do some of the things I was asked to do in strength, because I was a bit undervalued... and I have seen how I can trust myself. As well as the positive reinforcement from my coach, who constantly told me: “Yes, you can do it, yes you are capable! It has given me physical and mental strength.”*
(I.M.)


*“I must add that, apart from the satisfaction of exercising, the impact it had on the coach and the joy of, well, doing the exercises. Also, the fact of meeting faces that we usually know, but in other contexts, and seeing them there, was also a joy. I really enjoy socialising in a different but very satisfying space.”*
(J.T.)

## 4. Discussion

The aim of this research project was the implementation of the UAL Activa programme and its evaluation through qualitative research. Among the results obtained, the five blocks of content extracted from the voices of the participants stand out. In the first place, the improvements at a physical level are highlighted, where the improvement achieved in general terms is highlighted as a perception of greater agility and less physical fatigue. Closely related, we also find higher perceived values in their physical condition, highlighting flexibility, endurance, or strength. These findings would be in line with previous research showing how the practice of physical activity/exercise on a regular basis has reported physical improvements at various levels, not only physical condition but also at the level of the immune system [36,37,38].

Secondly, the physical activity programme seems to have had a positive impact on the emotional health of its participants, who report greater feelings of well-being, satisfaction, and vitality, as has been found in other similar research [39,40]. These findings highlight a greater ability to find emotional balance and greater harmony in the management of feelings and emotions in people who participate in physical activity and/or physical exercise as a healthy and active lifestyle habit [41].

Furthermore, it is evident that participation in the programme has led them to value their lifestyle, encouraging their adherence to daily physical activity and being an example of a healthy lifestyle for the people around them. These results are in line with past studies where regular practice over a certain period of time helps to internalise a lifestyle that becomes part of the personal routine and is valued both by the participants themselves and by their close environment that leads to imitating such adaptive behaviour patterns and habits to have and maintain an active life [42,43].

Finally, it should be noted that this physical activity is presented as a place for social encounters, which leads to the experience of social interactions and the creation of emotional ties with other people, which has adaptive consequences for people, as human beings are social beings by nature and part of their satisfaction or emotional well-being is subject to the possibility of relating socially with other people. In this way, thanks to the practice of physical activity, ideal scenarios are created to meet other people and establish social dynamics in a playful and relaxed context that leads to experiencing a pleasant climate full of good feelings for those who experience them [44,45].

There are already some studies that show the worrying data on the problem of obesity in our country and its consequences. Thus, the Organisation for Economic Co-operation and Development is insisting in its most recent reports that obesity is one of the most important risk factors that endanger health and that will have the greatest repercussions on the growth of health spending in the future. The OECD has highlighted that in the USA, health care costs are 36% higher and medication costs are 77% higher for obese people. These differences are also found in European countries. The growth of obesity over the past two decades in most OECD countries will translate into higher healthcare costs in the future. Its recent study on health expenditure projections devotes a whole chapter to obesity. If left unchecked, obesity may in the near future erode the gains in healthy longevity achieved today for older people, while adding to the burden of overall health costs [46].

The constant presence of the Nutrition and Physical Activity Strategy, also known as the NAOS Strategy, promoted by the Spanish Ministry of Health [47], in the media is another of its characteristics. Thus, one of its objectives, to raise public awareness of the health significance of the problem of obesity, is now closer, as the media in modern society is a fundamental channel for obtaining information and changing citizens’ behaviour. Therefore, intervention projects such as those carried out in this research would lead to continue adding in political-practical lines to the improvement and quality of social welfare.

## 5. Strengths and Limitations

The strength of the present study lies in the breadth of deployment achieved from the project implemented by UAL Activa in the university community and the high participation obtained for the collection of voices from a qualitative point of view. However, this same strength can become a limitation when it comes to replicating the study, due to the high level of participation, a work that was carried out over a long period of time, making data collection long and difficult. In addition, the lack of generalizability of the data and results to other populations would be another limitation to be taken into account.

## 6. Conclusions

Following the description of the programme and the users’ assessment of the initial objectives of UAL Activa, we would like to highlight the following conclusions:

First, there is a commitment to a healthy and active university through an original and innovative project, integrated in a transversal and participatory way throughout the university community of Almeria.

Second, a strategy for the promotion of physical activity for health has been established, offered by the Sports Unit through a new service offered, with the incorporation of final year students of the degree in physical activity and sport sciences specialised in healthy training, which means a practical training process that enables them to increase their professional qualifications and facilitates their future labour insertion more easily.

Third, the diagnosis, planning, programming, monitoring, and evaluation of the workers enrolled in the project has meant an important leap in quality for the individualisation of training according to personal characteristics.

Fourth, all UAL professionals have been offered the possibility of carrying out personal training with the advice and supervision of a future physical activity and sport professional expert in training, through the service-learning methodology.

Fifth, a real programme has been developed that has really favoured a healthy environment, with the purpose of adhering as many members of the university community as possible to an active and healthy lifestyle, in order to prevent a large number of illnesses and to carry out a strategy to promote the quality of life of the workers.

Sixth, an active methodology has been implemented which has favoured an innovative training process of exceptional quality for sport science students.

In short, UAL Activa has created in its participants a new way of life through the incorporation of physical activity into their lifestyle, which is reflected in their daily lives and in the way they act and live, from leisure time to work. Additionally, of course, we understand that this model of physical activation, taking into account human and material resources, is exportable to all types of public and private organisations.

## Figures and Tables

**Figure 1 ijerph-20-02350-f001:**
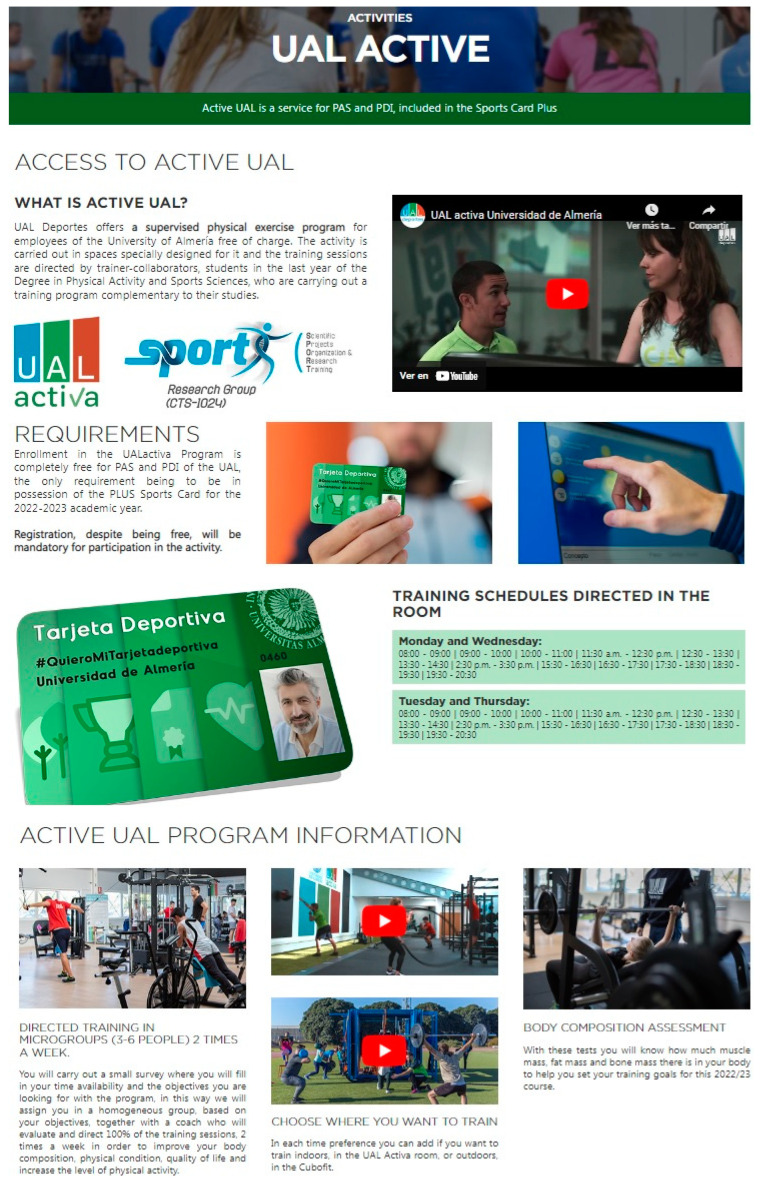
Information brochure of the UAL ACTIVA programme.

**Table 1 ijerph-20-02350-t001:** The categories and sub-categories of participant’s experiences.

Categories	Sub-Categories
Physical health	Physiological improvements Body pain Blood test Body composition
Fitness	Flexibility Resistance Strength Agility
Mood and emotions state	Psychological well-being Optimistic level Vitality feeling Relaxation sensations
Lifestyle values	Discipline Change in habits Healthier choices
Social interaction	Role of the coach Strengthening bonds with others Sense of belonging to a group Group motivation Family and friendly atmosphere

## Data Availability

Data presented in this study are available on reasonable request from the first author.

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
