# Peer review of "Promoting Physical Activity and Health in the Workplace: A Qualitative Study among University Workers, Spain"

_ijerph, 2023, doi:10.3390/ijerph20032350_

Round 1

Reviewer 1 Report

It has been a pleasure to read your work. Congratulations for all your deployment, organisation and study of this type of interventions in the population. I enjoyed reading it, well structured and guiding the reader every step of the way to the exposition of your magnificent project at the level of the institution of the University of Almeria, which I found fascinating. However, in order for your study to be fully complete I missed the inclusion of any barriers or limitations to this research. I would recommend including a brief description of these in the discussion section.

Author Response

Dear Reviewer 1,

First of all, thanks for your words towards the study. Secondly, thanks for your comment on the inclusion of a sub-section highlighting the difficulties and limitations of the study. A paragraph has been included in the new version taking this into account.  

Reviewer 2 Report

The manuscript “Promoting Physical Activity and Health in the Workplace” is significant for lifestyle promotion in the workplace. However, the manuscript needs major revision.

Add authors’ affiliations and Email below the name.

Title: add study design (a qualitative study) and setting (XXX).

Suggested title: Promoting Physical Activity and Health in the Workplace: A Qualitative Study among XXX in XXX, Country.

Abstract: please check the word count as per the journal requirement. Mention data analysis methods.

Introduction

Provide a comprehensive discussion. The rationale behind the study is not focusing. Remove subheading.

Move Figure 1. Information brochure of the UAL ACTIVA programme, to methods sections.

Methods

This section needs to be organised as

Study design

Context and study settings

Participants’ characteristics, participants’ recruitment strategies and ethical consideration

Data collection methods

Data management and analysis

Results

Add one Table on the coding tree – themes, categories, and codes

The results section contains more quotes. Please describe the content and provide selective quotes.

Discussion  

Need to be improved

Add one paragraph on Implication for policy and practice

How your results will be generalised to other context/settings.

Mentions strength and limitations of the study.  

Reviewer 3 Report

Conclusively, I suggest accepting this article after major revision. The reason is missing some necessary information for describing the UAL activa program.

Between the 218th - 221st lines of this manuscript, the authors stated that they implemented a semi-structured interview to ask the UAL active program's participants. I suggest the authors present this semi-structured interview in their manuscript.

Between the 237th to 324th lines, the authors summarized the results of a qualitative interview. I think the authors should provide more details of this qualitative interview if the conclusion of this article came from it. How to choose the participants of that qualitative interview? How to implement the qualitative interview? How to determine questions in the qualitative interview? I think readers of this article have similar questions; thus, they may generate similar studies.

In the Discussion, I suggest the authors provide their opinions after implementing their study. The present contents of this section seem to come from references. If possible, revise this Discussion section.

Round 2

Reviewer 2 Report

Thanks for addressing all the suggestions. 

Reviewer 3 Report

No further questions. The review can end.